# Barriers to Women’s Menstrual Hygiene Practices during Recurrent Disasters and Displacement: A Qualitative Study

**DOI:** 10.3390/ijerph21020153

**Published:** 2024-01-30

**Authors:** Shela Akbar Ali Hirani

**Affiliations:** Faculty of Nursing, University of Regina, 516 RIC, 3737 Wascana Parkway, Regina, SK S4S 0A2, Canada; shela.hirani@uregina.ca

**Keywords:** women, menstruation, disaster, displacement, hygiene, barriers, Pakistan

## Abstract

Disasters that involve displacement are particularly challenging for managing personal and menstrual hygiene, which can increase the risk of toxic shock syndrome, infections and other health conditions that can endanger women’s lives. This study aimed to examine the menstrual hygiene challenges experienced by internally displaced women affected by recurrent natural disasters and subsequent displacement in the context of a low–middle-income country, i.e., Pakistan. A critical ethnographic study was undertaken in disaster-relief camps in the northern region of Pakistan. Data were collected using multiple methods, including field observations, review of media reports and in-depth interviews with 18 women. The findings suggested that the key barriers to the personal and menstrual hygiene of women during recurrent disasters and displacement in the mountainous rural region of Pakistan include inadequate housing, lack of infrastructure and humanitarian aid, no waste disposal system and lack of women-friendly spaces in disaster-relief camps. Community-based collaboration is necessary for the implementation of effective interventions. A comprehensive menstrual response to promote the health and well-being of women during disasters must include menstruation supplies, supportive facilities (mainly toilets and bathing facilities), supplementary supplies for storing, washing and drying, disposal/waste management facilities, education and culturally appropriate spaces and supplies.

## 1. Introduction

Adequate menstrual hygiene is defined as clean absorbent supplies, privacy to change and wash, water and soap to wash, disposal facilities, education/awareness and dignity-affirming spaces [1,2,3,4]. As such, a comprehensive menstrual hygiene response, regardless of the disaster, must include menstruation supplies, supportive facilities (toilets, bathing), supplementary supplies (for storing, washing and drying), disposal/waste management facilities, education, as well as culturally appropriate supplies, messaging and problem solving [5,6]. There are many overlapping perspectives on the purpose of providing menstrual hygiene during disasters, including health and hygiene needs; privacy and safety, primarily against sexual violence; life-saving measures; dignity and empowerment; and education [6]. In the United Nations (1948) Universal Declaration of Human Rights [7], dignity is described as an inherent human right; however, without access to menstrual hygiene services, this is not possible [4].

Disasters that involve displacement are particularly challenging for managing menstrual hygiene [8,9]. Refugees, who have to travel as light as possible, may not be allowed to bring sanitary items. During rapid displacement, panicked packing means hygiene supplies may be forgotten, and after disasters, roads and shops are typically closed. Resource scarcity during disasters also means items such as hygiene products are often considered lower priority [1,10]. Not having access to opportunities to maintain adequate menstrual hygiene often results in bloodstained clothing and prolonged use of sanitary products, which can increase shame, embarrassment and the risk of toxic shock syndrome or other health conditions [1,11]. During disasters and displacements, when there is nowhere available to dry menstrual cloth privately, damp cloth is often reused, which increases the risk of rashes, fungal infections and urinary tract infections [12]. During disaster and displacement, when there is a shortage of menstrual supplies, women often resort to unhygienic alternatives out of necessity, which further increases the risk of infection [1,5,8,12,13,14]. A study reported that of over 800 women in refugee camps in Lebanon and Syria, almost sixty per cent did not have access to underwear, and even fewer had access to hygiene supplies [15]. Over fifty per cent of the participants experienced a urinary tract infection while at the camp, and many of them did not receive treatment [15].

In the context of low- and middle-income countries, women face add-on challenges in maintaining their menstrual hygiene during disasters due to a lack of infrastructure, mainly clean supplies and washrooms [15,16]. During disasters, evacuation centers, refugee camps and other temporary displacement shelters are often too crowded, inadequately supplied and lack bathing facilities [10,12,16]. As toilets and bathing facilities are commonly separate from temporary housing and shared by many individuals and families [5,16,17,18], many women often face violence and risks to their safety when leaving their informal settlements to go to the toilet [5]. In Myanmar, men are not prevented from using the women’s latrines, in addition to gaps in the walls and no locks on the doors [5]. Pathways are not lit at night, but many women wake up as early as 4 am to avoid the risk of intruders [5]. Syrian refugees living in Lebanon also had inadequate conditions, sharing cramped and dirty toilets with several families, with no privacy in the informal settlements, which are also not secure, and many fear violence when going to the toilet at night [5]. Rohingya refugees in Bangladesh also had fears of men peeping into the women’s latrines and bathing facilities, which did not have adequate privacy [18]. These findings suggest the need to examine a range of barriers surrounding the menstrual hygiene practices of women during disaster and displacement, especially in the context of low- and middle-income countries where women are at high risk of facing health concerns, exploitation and violence.

Pakistan is a low–middle-income South Asian country that has high mortality rates among women, especially in rural areas where there are a limited number of reproductive health services, healthcare facilities and trained healthcare providers [19]. Recurrent disasters (earthquakes and glacial lake outburst flooding) are common in the northern mountainous regions of rural Pakistan, resulting in the displaced communities experiencing homelessness and living in cramped living conditions, mainly in tents or shelters that lack basic facilities [20].

In Pakistan, where patriarchy is common and men are the primary decision makers in the determination and allocation of services/humanitarian aid, women are reported to face exploitation and abuse due to a lack of safe spaces during disasters [21,22,23,24,25,26]. Disaster-relief camps that are often far from cities often accommodate more than one family in a single tent/shelter and lack adequate privacy for women, which negatively affects women’s health and well-being [21,22,23,24,25,26]. Previous studies undertaken in post-disaster settings in Pakistan reported that displaced women in Pakistan often encounter food insecurity, oppression, violence, gender disparities and injustices during disasters, displacement and settlement in disaster-relief camps [21,22,23,24,25,26]. Internally displaced women in rural Pakistan are reported to have a high prevalence of reproductive infections due to improper menstrual hygiene practices [27]. These infections are the leading cause of rising mortalities and morbidities among Pakistani women [27]. To reduce the risks of reproductive infections leading to mortalities/morbidities among Pakistani women during disasters, there is a pressing need to explore key barriers shaping the menstrual hygiene practice of internally displaced women in disaster-relief camps in Pakistan.

Although approximately 65% of the Pakistani population lives in rural areas [24], there is a dearth of research on the menstrual hygiene-related barriers encountered by women in disaster-relief camps in Pakistan, especially in the context of rural mountainous regions where recurrent disasters are common and healthcare facilities are not readily available. It is essential to address this gap in knowledge to uncover the underlying barriers and identify possible solutions surrounding the menstrual hygiene practice of women facing recurrent disasters. Considering the risks associated with women’s health during times of disaster and the dearth of knowledge in this area, this study aimed to examine the key barriers to the menstrual hygiene practices of internally displaced women residing in the disaster-relief camps of an underserved mountainous region of Pakistan.

## 2. Materials and Methods

A critical ethnographic study was undertaken in the disaster-relief camps in the northern region of rural Pakistan, where thousands of disaster-affected families were residing in temporary shelters, tents and camps located at the top of the mountain. Critical ethnography examines struggles, power relationships and conflicts in the lives of vulnerable and marginalized populations living in a specific cultural context [28,29]. The chosen study design provided an opportunity to uncover menstrual health disparities in the lives of women who were forced to live in temporary shelters/tents after recurrent disasters and analyze gaps in the offered services. The critical nature of this inquiry uncovered hidden realities and gave voice to the marginalized women affected by recurrent disasters in the remote mountainous region of Pakistan. Moreover, it facilitated the researcher to examine key barriers in women’s social structures that are detrimental to their menstrual hygiene practices. The researcher approached this critical ethnographic research as a partial insider and partial outsider. The principal investigator is a Pakistani–Canadian woman who is a partial insider as she is fluent in Urdu (the national language of Pakistan) and has resided in Pakistan. On the other hand, the researcher is a partial outsider because she is an educated woman who has no prior experience living in the disaster-relief camps of the rural villages of a low–middle-income country. The critical nature of the inquiry enabled the researcher to remain mindful of the power differences throughout fieldwork and during the iterative process of data collection and analysis, hence carrying forward the voices of marginalized women residing in tents/shelters during recurrent disasters.

Fieldwork was undertaken in four disaster-affected villages of Chitral, Pakistan, with the support of a humanitarian relief agency based in Pakistan. Chitral is a remote mountainous region that has more than 5000 glaciers, which makes this place prone to a variety of recurrent natural disasters, including flash flooding, land sliding, glacial lake outburst flooding and avalanches. This geographic region still experiences recurrent earthquakes of mild to moderate intensity. The subsequent and recurrent disaster affected the community and contributed to long stays in tents or shelters, more than 2.5 years. After receiving approval from the humanitarian relief agency and ethics review board, the data were collected by the principal investigator who travelled to the disaster-relief camps in the mountainous region of Chitral during the active phase of disaster (earthquake) in the region.

Data were gathered using multiple methods, including field observations, reviews of documents maintained by the humanitarian relief agency and in-depth interviews with 18 women after seeking informed consent.

Field observations were specific to the availability of menstrual services, supplies and environmental resources available for women affected by disaster and displacement. The principal investigator observed the washrooms and bathing facilities available to disaster-affected communities, as well as noting their level of cleanliness and distance from the temporary housing/shelter. The principal investigator also made observations on the availability of water supply and sanitation facilities in the disaster-affected villages. Furthermore, the researcher observed the layouts of the tents/shelters and the nature of humanitarian aid offered to the disaster-affected community, especially women. Field observations were also made on the availability of menstrual products in the community, hygiene products commonly used during menstruation and ways these products are disposed of. The researcher kept field notes of all the observations made during the fieldwork.

The principal investigator also reviewed documents, including an informal report maintained by the humanitarian relief agency and three newsletters published by the health services department in the region. These documents were reviewed to gather add-on information on types of healthcare and social services offered to internally displaced women, identify disparities in the humanitarian aid targeting women and recognize gaps in the menstrual hygiene services or facilities for women during recurrent disasters and displacement. Field notes were made while reviewing these documents.

For in-depth interviews, participants were recruited using the purposive sampling method. Adult women aged 18 years and above were invited to participate in the study with the help of a local community mobilizer who was familiar with the community’s cultural norms and local languages. In total, 18 women who accepted the invitation and provided informed consent were interviewed by the principal investigator for 45–60 min using a semi-structured interview guide to explore the menstrual hygiene challenges of women during recurrent disasters. The interview guide had probes to explore environmental challenges, personal experiences and recommendations to overcome menstrual hygiene-related challenges for internally displaced women. These interviews were undertaken in Urdu (the national language of Pakistan). All participants were interviewed in their temporary housing (shelters/tents) while assuring the comfort and privacy of the participants. All the interviews were audio-recorded, transcribed and then translated into English. With the help of a language expert, an audit trail of these interviews was undertaken to verify the accuracy of the translation. All participants were told about the voluntary nature of their participation and were provided an option to withdraw from the study by contacting the principal investigator within one month of the fieldwork. Each participant was assigned an identification number to assure their confidentiality during each step of the research process.

Data from this critical ethnographic study were analyzed inductively and iteratively. The data gathered through in-depth interviews, field observations and document analysis in the form of transcripts and fieldnotes were analyzed manually by the principal investigator. A community-based representative was also involved in data analysis who assisted the researcher in paying close attention to the context and critically examining data in relation to the wider environmental factors. This approach facilitated the examination of disparities, barriers in social structures and gaps in services affecting the menstrual hygiene practices of internally displaced women in Chitral, Pakistan. Multiple steps were followed to analyze data gathered through multiple sources. The first step involved the selection and isolation of the codes, depicting factors that directly and indirectly affect the menstrual hygiene practices of internally displaced women. The next step involved comparison and validation of codes derived from in-depth interviews, document reviews and field observations. The subsequent step involved the identification of categories by paying close attention to the range of environmental factors affecting the menstrual hygiene practices of women who were residing in temporary shelters/tents after recurrent disasters and displacement. As a final step, four broader themes were derived that represent menstrual hygiene challenges that require need-based services for women residing in the context of disaster-relief camps. To assure the trustworthiness of the analyzed data, participants were contacted to seek verification of the interpretation drawn from the data gathered through in-depth interviews, review of documents and field observations. Triangulation of data gathered through multiple methods was another strategy that facilitated the trustworthiness of the gathered information and helped in gaining in-depth insight into the phenomenon.

## 3. Results

### 3.1. Characteristics of Study Participants

Study participants were residing in the disaster-relief camps of Chitral, Pakistan, after the glacial lake outburst flooding and recurrent earthquakes. The age of participants ranged from 18 to 40 years, and their education level ranged from illiterate to post-secondary education. Out of 18 participants, 10 women were able to read and write their native and national languages, mainly Khowar and Urdu. The majority of these participants were Muslim and used a strict veil system, as per their cultural and religious norms. In Chitral, where early marriages at the time of puberty are common, all the participants were married and had one to seven children. Out of 18 women, 14 participants were living in extended families. The total number of family members living in a single tent/shelter ranged between 4 and 15 members. Their demographic characteristics are summarized in Table 1.

### 3.2. Barriers to Menstrual Hygiene during Disasters

Four themes were derived to present the barriers surrounding women’s menstrual hygiene practices during recurrent disasters and displacement in the mountainous rural region of Pakistan. These barriers include inadequate housing, inadequate infrastructure and humanitarian aid, no waste disposal system and lack of women-friendly spaces. Each of these themes is discussed below.

#### 3.2.1. Inadequate Housing

Following a disaster, participants had to live in the mountains, in relatives’ houses, their own damaged house or in a tent or shelter allocated by the humanitarian relief agency; therefore, inadequate housing served as a key barrier to their personal and menstrual hygiene. All the participants shared the struggles surrounding living in a shelter after recurrent disasters that destroyed their houses and agricultural lands. Almost all the participants shared that they use clothes during menstruation and have no space in their tents/shelters to change, wash or dry these menstrual clothes.

Field observations and reports maintained by the humanitarian relief agency revealed that the recurrent disasters accumulated a large amount of debris on both the residential and agricultural land, due to which there was no land available to rebuild houses/place shelters or tents at a safer location that has access to a water supply and washroom facilities. During the fieldwork, the researcher observed that the participants were living at the top of the mountain/hills in a single tent/shelter with limited space in relation to family size. These overcrowded tents and shelters were not weatherproof, had no bathing facilities and water supply and there was no space for women to wash or dry their menstrual clothes. In many villages, women had to travel downhill to the river to wash their menstrual clothes, drying these clothes by hiding them in plastic bags.

During in-depth interviews, women shared that it is challenging to maintain their personal and menstrual hygiene as they do not have washrooms, bathing facilities, a water supply, privacy or space to clean menstruation clothes in their temporary housing (shelters and tents). Women felt uncomfortable attending to their personal and menstrual hygiene in overcrowded shelters and tents shared with male members of the family. Many mothers opted to defecate and urinate in fields that made them feel uncomfortable and stressed. They further shared that due to inadequate housing, they feel unsafe, insecure and embarrassed to go to communal washrooms that are located in distant places and shared by everyone. While sharing the challenges surrounding personal and menstrual hygiene, one participant shared.

The tents which we were given were not that comfortable or spacious for us as there were more people in each family and that is why we had difficulties there. We would all go to a space in an open area and urinate there if we needed to. We didn’t have anything in the tents. We would go there in the night. The damaged washrooms which were spared in the floods were still useful so during the day we would use them. Everyone [men and women] came down to use those washrooms. During the day we used to climb down from the mountain to use the damaged washrooms. But we would go out in an open area somewhere during the night. We would take our men with us because the place was deserted, anything could happen there, God forbid. We could have had a fall or there could be wild dogs there too, but during the day we would climb down from the mountains to use those damaged bathrooms. By the time we used to reach these places to urinate, we had a feeling our urine had dried up.

Another participant living in a small-sized shelter with a large family shared challenges surrounding maintaining her menstrual hygiene while living in a shelter that had no place to change, wash or dry menstrual clothes.

Women feel very uncomfortable during menstruation and they also have stomach aches. They cannot sit with the men for long, so they can easily lie down in there. During the rainy days, we couldn’t go down to the damaged washrooms to change the menstruation clothes, so we placed a curtain in the tent. The women who had their menses would sleep on the other side of the curtain while the men slept on one side. We would dry the washed cloth in the middle of the night and would use the other ones till then. We kept them under trees because there was no other place, or we kept them hidden under our beds to dry the menstruation clothes.

#### 3.2.2. Inadequate Infrastructure and Humanitarian Aid

Participants highlighted inadequate infrastructure and humanitarian aid as another key barrier surrounding their menstrual hygiene as there was a non-availability of menstrual supplies, safe spaces or any bathing, cleaning and washing facilities in the disaster-relief camps. A review of documents indicated that during the time of disaster, humanitarian aid mainly comprised tents/shelters, food items and financial assistance for a limited time (maximum one month), as well as non-food items, like basic utensils, blankets and mattresses. There was no mention of the provision of hygiene products or related support/education for menstruating women during disasters and displacement. During fieldwork, the researcher observed that menstrual products were not commonly available in many villages as disaster led to the destruction of infrastructure, and medical/general stores were no longer operational in that region.

During in-depth interviews, women shared that as they had no financial resources or access to menstrual supplies through humanitarian aid, they were using their children’s clothes as their menstrual pads. Many were unable to change these blood-stained menstrual clothes for more than 2 days as communal washrooms had no water supply or space to clean, wash or dry them.

Displaced women shared that despite being aware of the negative effects of poor menstrual hygiene practices, they were unable to take care of their hygiene, especially during the cold weather. Field observations supported that Chitral is a northern mountainous region of Pakistan that experiences severe weather conditions and freezing temperatures ranging between −7 and −15 C. In-depth interviews with participants underscored that it was challenging to take care of their hygiene in the absence of required and need-based supplies, facilities and a warm water supply in cold-weather conditions.

During fieldwork, it was observed that there were no separate and private bathing facilities for displaced women within 2 km. As there were no washing facilities, space to dry clothes (especially women’s undergarments and menstruation clothes) or water supply in the washroom attached to the shelter, women had to go down a hill discretely to wash their menstrual clothes in a canal and river in the early morning. While sharing the challenges surrounding the lack of infrastructure and humanitarian aid, one of the participants shared

If we recall that time we feel like crying. At that time, the clothes that we wore were also dirty and smelled, there was no place where we could wash them and there was nowhere to go because we couldn’t leave our children while they slept as it was a deserted area.

#### 3.2.3. No Waste Disposal System

Another barrier that negatively affected the menstrual hygiene practices of displaced women in disaster-relief camps in rural Pakistan was the non-availability of a waste disposal system. During in-depth interviews, women mentioned that during the recurrent disasters and displacement from one village to another, it was impossible to wash and dry their used menstrual clothes.

As there was no waste disposal system available in disaster-relief camps, many women discretely buried or burned their menstrual clothes on the ground after using those damp clothes for several days. One of the participants shared the challenge of disposing of her menstruation clothes. She stated:

We don’t have a gutter line [sewerage system]. During menses, we looked after ourselves and threw our clothes as well which we used. We dig up the ground and put our menstrual clothes inside the ground

During field observations, it was noticed that there was no waste disposal system in the villages. Local villagers tried to dispose of their waste by dumping or burning it. Disputes were observed among villagers and mothers of young children who believed that the improper disposal of waste by their neighbors was accumulating flies outside their shelter/mud-brick house and affecting their children’s health. A review of documents also highlighted the lack of waste disposal as a key challenge in the region that is increasing risks of skin infections and other health issues.

#### 3.2.4. Lack of Women-Friendly Spaces

An additional barrier that hindered the menstrual hygiene practices of women during disasters was the unavailability of women-friendly spaces in the setting of disaster relief that can offer privacy to women. During fieldwork, the researcher observed that women were living in cramped spaces, had exclusive spaces to take care of their hygiene, were expected to seek permission from a male family member before leaving their tents/shelter and were expected to fully cover themselves considering their religious/cultural norms. Hence, they were experiencing power, control and risks toward their health in terms of a lack of personal hygiene. A review of documents further confirmed that the majority of the internally displaced women in that region were of child-bearing age, among whom a rise in mortality and morbidity was reported during and after childbirth.

Participants wished for gender-sensitive culturally appropriate spaces for women that are clean, accessible and exclusively for women where they can take care of their hygiene without any intrusion from male members of the community. A participant shared their challenge related to the unavailability of a gender-sensitive and culturally appropriate space to maintain their hygiene, especially during menstruation and the postnatal period:

Women who are menstruating have difficulty because they don’t have a separate room where they can dry the used clothes and wash them as well. There is a special place needed for this. We cannot dry them in the bathrooms or anywhere outside. It becomes very difficult for us. That is why I am saying that they should make some proper arrangements for us. It is very troubling. When a woman gives birth to a child she needs to stay in a shelter and stay somewhere for almost a month. She also gets her period at that time. There is only one washroom which she cannot use properly because others are using that too. That is why it is troubling. You cannot dry these clothes [menstruation clothes] in the washroom or anywhere outside.

Another participant who could not take care of her hygiene after her childbirth and was living in an overcrowded shelter recommended:

They [women] should also have room to take a bath because it gets extremely difficult during the winter and snow. Taking showers is chaos both for women and children. There should be a separate room for the ladies or at least one bathroom allocated to 5–6 houses where they can go and wash their clothes, wash their hair and give showers to their children with warm water. So, mothers should be given a proper bathroom [Hamaam is the local term] with a warm water supply.

While sharing the challenges and inconveniences during the period of menstruation, another participant recommended:

The washroom should be big enough and there should be some facilities where it gets easier for the women to wash their mensuration clothes and hang them in the washroom for drying. There should be some heating units for the women during the winter where they can place these clothes and they can dry them because there is no space outside.

## 4. Discussion

This study indicated that in the context of low–middle-income countries, where recurrent disasters are common, some of the key barriers to menstrual hygiene include inadequate housing, inadequate infrastructure and humanitarian aid, no waste disposal system and a lack of women-friendly spaces in disaster-relief camps. The findings suggested the importance of a cultural- and gender-sensitive approach in providing need-based care to the vulnerable group of women who face hardships in ensuring their menstrual hygiene practices during times of disaster and displacement due to a lack of adequate menstrual supplies, bathing spaces, water and disposal systems. These aspects are essential to protect displaced women from a variety of infections. Insufficient bathing and laundry facilities increase the chance of rashes, skin infections, urinary tract infections and other health risks [12,14,17].

There is an increased demand for water during menstruation (for bathing, washing clothes and menstrual supplies). However, displacement settings commonly have scarce water supplies, and women may be afraid that men will see their use of more water as unfair [10]. Similar to this study’s findings, there was no convenient access to water in most latrines and washrooms in low–middle-income countries like Myanmar or Lebanon, contributing to the difficulties women in these refugee camps faced [5]. Access to water and sanitation is a human right, recognized by all United Nations Member states in 2015 [30,31]. Therefore, providing sufficient water to enable adequate menstrual health management is a crucial consideration for any disaster-relief organization. When adequate water is not available, supplies need to shift accordingly to include items such as wet wipes and larger quantities of disposable menstrual products [5,10].

This study highlighted that women need private and safe spaces to ensure their mensural hygiene during times of disaster and displacement. Privacy is a key component to ensuring that menstruation can be managed with dignity, hygiene and safety [6,12]. Private spaces are necessary for many reasons, including areas to change menstrual products, bathe, wash clothes and other reusable items, as well as somewhere to dry them [5,12,16]. Women in India, Myanmar, Bangladesh and Lebanon all shared concerns about having no privacy in the bathing facilities provided in disaster-relief camps [5,17,18]. In Malawi, laundry bags were used to add privacy for young refugees washing and drying their menstrual clothes, designed with frequent consultation and collaboration to prioritize their needs [32]. Unlike the current push for gender-neutral bathrooms in many high-income countries, the literature stresses the importance of providing gendered washing and hygiene facilities in separate locations for refugee camps and other temporary displacement shelters, for both privacy and safety [5,6,10]. Additionally, these facilities should have solid doors with locks and be equipped with garbage bins, soap, good lighting, space and ventilation [10,11,15].

In the context of low- and middle-income countries, the role of education and guidance on the maintenance of menstrual hygiene is crucial to empower women and prevent health risks. Education is primarily recommended for those who access menstrual hygiene services, volunteers and staff responsible for distributing menstrual hygiene supplies [3]. Education is also a primary way to empower girls and women [31,33]. Education recommended for refugees and other disaster-affected individuals is often aimed at puberty guidance and awareness because so many participants do not know about menstruation before it happens to them. However, when young people are educated about their bodies, they are less likely to be afraid or anxious about things like menstruation [33]. Counselling services are also recommended to help women with the psychological effects of menstruation, particularly adolescents getting their first cycle [14]. Efforts to normalize talking about menstruation without shame (e.g., community health education and social support) are necessary for it to be properly prepared for, and this benefits communities both pre- and post-disaster [3,14,16,17,34,35]. Practitioners involved with menstrual hygiene planning, programming and advocacy must be trained in the assessment and delivery of services in culturally appropriate ways. Moreover, they need to have a holistic and intersectional understanding of the associated challenges, including stigma, discrimination and human rights violations [31]. Another aspect of education is the use and disposal of unfamiliar menstruation products. Participants in multiple articles describe receiving unfamiliar supplies and inadequate practical information on how they are used, including disposable pads and reusable pads or cups [5,10,16,17]. There also needs to be clear signage and communication of where health and education facilities are within relief camps or refugee settlements and what services they offer. Most participants in the Ugandan refugee camp were unaware that there was a health center available to them [14].

The role of humanitarian aid targeting the distribution of hygiene kits and the creation of separate washrooms for females to have disposal facilities is crucial during times of disaster and displacement. Similar to this study, participants in Vanuatu wished for a separate menstrual hygiene distribution center staffed with women and men who would act professionally, so they could avoid embarrassment when picking up other supplies from distribution centers [10]. Disaster relief organized by Action Aid usually includes women’s safe spaces, which serve a variety of purposes including the distribution of menstrual hygiene kits [1]. Rohingya participants in Bangladesh wished for multifunctioning bathrooms so that it was less obvious when someone was accessing the facilities due to menstruation. With this feedback, OXFAM was able to construct a communal female WASH unit, including private toilets, showers and laundering areas [18]. In some of the female WASH units, an innovative menstrual disposal system was piloted, where a “discreet chute system” connected to a specific container treated to store the waste [18] (p. 7). In this context, NGOs and disaster relief workers had to collaborate with men as well to create an efficacious system for the Rohingya girls and women, because their mobility is strictly controlled by the male leader of their family unit [18]. Different age groups may also have different preferences and practices [12], and the wishes of younger demographics are often less considered [6,14]. The NGO Plan International created laundry bags to give girls at a refugee camp in Malawi privacy for washing and drying their menstrual products [32]. These bags were made with frequent consultation and collaboration with the girls at this camp until they fit all their needs and priorities, using a “human-centred design” [32] (p. 29). This initiative differs from most others that have engaged with community collaboration because the participants were involved with the development process as well, rather than starting with preconceived ideas [32].

Community-based collaboration is necessary for the implementation of innovative, need-based and effective menstrual hygiene services and disposal facilities during disasters and displacement. Community-based participatory assessments and consultation with the people accessing these services and menstrual supplies are also necessary to ensure these services/supplies are sustainable and appropriate for each context [3,6,11,36,37]. People who menstruate need to be involved in emergency response planning before, during and after disasters [10], and when relief organizations come into new contexts, there needs to be a consultation process so local needs and preferences can be considered. When the preferred menstrual supplies are not known ahead of time, many relief organizations distribute whatever supplies are readily available first and then adjust as they consult recipients [6]. However, once local preferences are understood, supplies can be bought in bulk to prepare for future disasters [11].

The findings suggest that regions that regularly experience recurrent disasters need to make preparation and planning a priority. A common challenge impeding menstrual hygiene-related services during disasters is confusion around which sector is responsible for its preparation and distribution, as it includes water, sanitation, waste management and public health. Coordination among these different sectors is crucial for providing a holistic systemic response during emergencies [3,5,6,14,16,17]. Funding must be provided and identified for all components of menstrual hygiene services across the relevant sectors before emergencies to avoid assumptions and confusion [6]. Interviews with local emergency managers in Kansas demonstrated that when they were not surprised by questions about menstruation, they assumed partnering organizations would be responsible for those supplies, indicating that “unless emergency response agencies are specifically tasked with addressing menstrual hygiene needs, it is reasonable to expect that they will not” [8] (p. 60). Ensuring that women are included in the planning and execution of emergency aid is an important measure for providing useful aid. The emergency teams at Action Aid are led by women to help make sure that “the needs of women, refugees and other vulnerable groups are met” [1]. Another recommendation to help relief organizations prepare for disasters around the world, which does not appear to have come to fruition yet, involves mapping the menstrual practices, beliefs and preferred materials of different countries, regions and cultural groups around the world [6]. Continued research into disaster relief in different contexts is essential for improving menstrual hygiene practices, services and emergency responses in the context of low- and middle-income countries.

## 5. Conclusions

Recurrent disasters that involve displacement make it challenging for women to maintain adequate menstrual hygiene, especially in the context of low- and middle-income countries like Pakistan. A critical ethnography in the underserved mountainous region of Chitral, Pakistan, indicated that women face many challenges in maintaining their menstrual hygiene during disaster and displacement due to inadequate housing, the unavailability of separate/private bathing facilities, unavailability of a gender-sensitive and culturally appropriate space, no waste disposal system and a lack of women-friendly housing in disaster-relief camps. The findings suggest that community-based collaboration, participatory assessments and consultations with those accessing menstrual services are necessary for the implementation of effective and need-based menstrual hygiene services and interventions during recurrent disasters. A comprehensive menstrual response to promote the health and well-being of women during disasters must include menstruation supplies, supportive facilities (mainly toilets and bathing facilities for women), supplementary supplies (for storing, washing and drying), disposal/waste management facilities, education and culturally appropriate spaces and supplies. Women affected by disasters must be empowered through education and included in the planning and execution of emergency humanitarian aid and menstrual hygiene kits to ensure that these are need-based, context-specific and timely. Future research testing the effectiveness of these interventions is recommended to safeguard the health and well-being of displaced women during disasters and displacement.

## Figures and Tables

**Table 1 ijerph-21-00153-t001:** Characteristics of study participants.

Characteristics	Findings
Women’s Age	18 to 40 years
Education	
Illiterate	8
Grades 1 to 10	6
Bachelor’s degree	4
Marital Status	
Married	18
Unmarried/Single/Divorced/Widowed	0
Literacy	
Can read and write native and national language	10
Can’t read and write native and national language	8
Religion	
Muslim	17
Kalashi	1
Family Type	
Extended	14
Nuclear	4
Housing Type	
Shelter	14
Tent	3
Unstable house	1

## Data Availability

The data contain sensitive information about study participants and, based on the informed consent process, data containing participant information cannot be shared publicly.

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
