# Peer review of "Barriers to Women’s Menstrual Hygiene Practices during Recurrent Disasters and Displacement: A Qualitative Study"

_ijerph, 2024, doi:10.3390/ijerph21020153_

Round 1

Reviewer 1 Report

Comments and Suggestions for Authors

The manuscript as a whole is clearly written and presents a thoughtful narrative. The research objective is well-stated in the introduction and subsequently fulfilled in the results and discussion. The topic, while somewhat niche, is a valuable contribution to the literature, as menstrual hygiene in disaster settings is understudied compared to other contexts.

However, I find the reporting of the results to be inadequate. See additional specific comments below. I am happy to review a revised version of this manuscript in which the methods are more thoroughly described. 

Specific comments:

The author states that “data was gathered using multiple methods, including field observations, reviews of documents maintained by the humanitarian relief agency, and in-depth interviews with women…” However, none of the descriptions of these methods are sufficiently detailed. In the case of field observations and document review, each is afforded only a single sentence of description. For observations, what specific indicators were observed? Where? What settings/demographics were observed? What was the sample size? For document review, what specific documents were reviewed? How many? What information was extracted from them and how? Interviews are somewhat more thoroughly described but still do not report key items. For example, what topics were included in the interview guide?

“The process of data analysis involved the derivation of the codes.” How? Were any codes identified a priori, or were all codes derived inductively from reading the transcripts? If the former, what theory informed deductive codes? If the latter, how were codes developed and refined inductively across round(s) of coding? Were texts of news articles and reports coded in the same way as interview transcripts? How were observations coded (and was there even text-based data from observations)?

I recommend the author adhere to reporting guidelines for the relevant study types. Certainly reporting guidelines for qualitative research (e.g., SQRQ https://www.equator-network.org/reporting-guidelines/srqr/) apply. Other guidelines related to reviews may also apply for the document review component.

If the author prefers to keep the manuscript more succinct, providing a supplemental file may be appropriate for some details, though additional key information is needed in the main body of the manuscript even if the author chooses to add a supplement.

This may be related to clarity of methods reporting, but the results appear to present exclusively findings from interviews. Data from observations/document review is not explicitly presented. Were these data used and analyzed beyond triangulation of interviews?

Finally, the author states that a locally-based community representative co-analyzed the data. Does this individual not meet authorship criteria, or why are they not included?

Author Response

Response to Reviewer 1

Thanks for the appreciation and for sharing your valuable comments. All the suggested revisions are incorporated. Here are responses to the comments

Comment # 1

The author states that “data was gathered using multiple methods, including field observations, reviews of documents maintained by the humanitarian relief agency, and in-depth interviews with women…” However, none of the descriptions of these methods are sufficiently detailed. In the case of field observations and document review, each is afforded only a single sentence of description. For observations, what specific indicators were observed? Where? What settings/demographics were observed? What was the sample size? For document review, what specific documents were reviewed? How many? What information was extracted from them and how? Interviews are somewhat more thoroughly described but still do not report key items. For example, what topics were included in the interview guide?

Response to Comment # 1: As suggested, all the available information on field observations, document review and in-depth interviews are added. [Refer to Material and Methods Section on page 4, lines 146 to 182]

Comment # 2

“The process of data analysis involved the derivation of the codes.” How? Were any codes identified a priori, or were all codes derived inductively from reading the transcripts? If the former, what theory informed deductive codes? If the latter, how were codes developed and refined inductively across round(s) of coding? Were texts of news articles and reports coded in the same way as interview transcripts? How were observations coded (and was there even text-based data from observations)?

Response to Comment # 2: Steps on data analysis and the process undertaken for the derivation of themes are also elaborated. [Refer to Material and Methods Section on pages 4 and 5, lines 183 to 200].

Comment # 3

I recommend the author adhere to reporting guidelines for the relevant study types. Certainly reporting guidelines for qualitative research (e.g., SQRQ https://www.equator-network.org/reporting-guidelines/srqr/) apply. Other guidelines related to reviews may also apply for the document review component.

Response to Comment # 3: As advised, as per the reporting guideline of the qualitative study, all the steps involved in the research process are elaborated in the material and methods section. [Refer to Material and Methods Section on pages 4 and 5]. Newly added information in the manuscript is highlighted in red font.

Comment # 4:

If the author prefers to keep the manuscript more succinct, providing a supplemental file may be appropriate for some details, though additional key information is needed in the main body of the manuscript even if the author chooses to add a supplement.

Response to Comment # 4: As suggested, add-on information is added to the revised draft of this manuscript. Newly added information in the revised draft of manuscript is highlighted in red font.

Comment # 5

This may be related to clarity of methods reporting, but the results appear to present exclusively findings from interviews. Data from observations/document review is not explicitly presented. Were these data used and analyzed beyond triangulation of interviews?

Response to Comment # 5: Information gathered through in-depth interviews, field observations and document review has now been added in the findings section. [Refer to the results section, pages 6-9. Newly added information is highlighted in red font].

Comment # 6

Finally, the author states that a locally-based community representative co-analyzed the data. Does this individual not meet authorship criteria, or why are they not included?

Response to Comment # 6: A community-based representative who participated in data analysis does not fully meet the authorship requirements. Although the contribution of that individual is acknowledged in the acknowledgement section, the identity of that individual is kept confidential as per their request.

Reviewer 2 Report

Comments and Suggestions for Authors

Overall, the manuscript is well-written, and I have a few suggestions to make the manuscript more comprehensible:

In the Introduction section, on page 2, line number 54, it mentions, "A study reported that of over 800 women in refugee camps in Lebanon and Syria, almost sixty per cent did not have access to underwear, and even fewer had access to hygiene supplies." Could you please cite this particular study here?

Could you please explain the meaning of "due to a lack of infrastructure"?

Please provide the citation for the statement: "In Myanmar, men are not prevented from using the women’s latrines, in addition to gaps in the walls and no locks on the doors."

The study needs more explanation on the "barrier factors of women’s menstrual hygiene practices" in disasters and displacement settings. This is the central concern.

Please provide more context for conducting this research in Pakistan. The current explanation of the context of Pakistan (the last paragraph of the Introduction section) is inadequate.

In the Method section, the author has mentioned "Using a critical lens" for the reviews; we wonder what a "critical lens" is and need more elaboration.

Please explain how the data was analyzed.

Please provide a table demonstrating the study sample’s characteristics.

Explain how you derived the "four themes." Why not five themes? What is the method, and what is the procedure?

The themes explored in the study are generally imagined as we merely examine the article's topic. However, as we consider this, how did the author "use a critical lens" to interpret or explore the themes of the study? Please also clearly state how the authors employed "a critical ethnography" in the current study. These concerns need to be addressed in the manuscript.

Author Response

Response to Reviewer 2

Comment # 1

Overall, the manuscript is well-written, and I have a few suggestions to make the manuscript more comprehensible:

Response to Comment # 1: Thanks for the appreciation and for sharing your valuable comments. All the suggested revisions are incorporated. Here are responses to the comments

Comment # 2:

In the Introduction section, on page 2, line number 54, it mentions, "A study reported that of over 800 women in refugee camps in Lebanon and Syria, almost sixty per cent did not have access to underwear, and even fewer had access to hygiene supplies." Could you please cite this particular study here?

Response to Comment # 2: Citation is added to “A study reported that …. hygiene supplies” [Refer to page 2, line 55].

Comment # 3:

Could you please explain the meaning of "due to a lack of infrastructure"?

Response to Comment # 3: Information on “lack of infrastructure” is elaborated [Refer to page 2, line 60]

Comment # 4:

Please provide the citation for the statement: "In Myanmar, men are not prevented from using the women’s latrines, in addition to gaps in the walls and no locks on the doors."

Response to Comment # 4: Citation is added to the statement “In Myanmar, men are not prevented…on the doors” [Refer to page 2, line 67].

Comment # 5:

The study needs more explanation on the "barrier factors of women’s menstrual hygiene practices" in disasters and displacement settings. This is the central concern.

Response to Comment # 5: This aspect is elaborated. The background section initially presents the general barriers towards women’s menstrual hygiene practices during disaster and displacement settings. This section further narrows down to the context of low-middle income country and presents rationale for conducting this study in the rural mountainous region of Pakistan (a low-middle-income country) where recurrent disasters are comment. [Refer to the introduction section, pages 2-3, lines 40-109]. In accordance to the research aim of this study, the results section shares findings on barriers towards women’s menstrual hygiene practices during recurrent disasters and displacement. [Refer to the results section, page 5-9].

Comment # 6:

Please provide more context for conducting this research in Pakistan. The current explanation of the context of Pakistan (the last paragraph of the Introduction section) is inadequate.

Response to Comment # 6: The explanation of the context and rationale for conducting this study in Pakistan is elaborated. See the introduction section [pages 2-3, lines 58-109].

Comment # 7:

In the Method section, the author has mentioned "Using a critical lens" for the reviews; we wonder what a "critical lens" is and need more elaboration.

Response to Comment # 7: In the method section, the use of a critical lens in critical ethnography is elaborated. [Refer to Material and Methods Section on page 3, lines 111 to 131].

Comment # 8:

Please explain how the data was analyzed.

Response to Comment # 8: Steps on data analysis are also elaborated. [Refer to Material and Methods Section on pages 4 and 5, lines 183 to 200].

Comment #9:

Please provide a table demonstrating the study sample’s characteristics.

Response to Comment # 9: As advised, a table containing demographic information on study participants is added [Refer to page 5].

Comment # 10:

Explain how you derived the "four themes." Why not five themes? What is the method, and what is the procedure?

Response to Comment # 10: Steps on data analysis and the process undertaken for the derivation of themes are also elaborated. [Refer to Material and Methods Section on pages 4 and 5, lines 183 to 200].

Comment # 11:

The themes explored in the study are generally imagined as we merely examine the article's topic. However, as we consider this, how did the author "use a critical lens" to interpret or explore the themes of the study? Please also clearly state how the authors employed "a critical ethnography" in the current study. These concerns need to be addressed in the manuscript.

Response to Comment # 11: Information on data analysis using a critical lens is elaborated [Refer to Material and Methods Section on pages 4 and 5, lines 183 to 200]. Also, the rationale for using critical ethnography is elaborated [Refer to Material and Methods Section on page 3, lines 111 to 131].

Round 2

Reviewer 1 Report

Comments and Suggestions for Authors

I find that the author's revisions have meaningfully improved this paper and recommend it be accepted for publication.

Reviewer 2 Report

Comments and Suggestions for Authors

The revised version is much better than the initial one, and it is now acceptable.